# Anaplastic Lymphoma Kinase (ALK) Inhibitors Enhance Phagocytosis Induced by CD47 Blockade in Sensitive and Resistant ALK-Driven Malignancies

**DOI:** 10.3390/biomedicines12122819

**Published:** 2024-12-12

**Authors:** Federica Malighetti, Matteo Villa, Mario Mauri, Simone Piane, Valentina Crippa, Ilaria Crespiatico, Federica Cocito, Elisa Bossi, Carolina Steidl, Ivan Civettini, Chiara Scollo, Daniele Ramazzotti, Carlo Gambacorti-Passerini, Rocco Piazza, Luca Mologni, Andrea Aroldi

**Affiliations:** 1Department of Medicine and Surgery, University of Milano-Bicocca, 20900 Monza, Italy; f.malighetti@campus.unimib.it (F.M.); m.villa96@campus.unimib.it (M.V.); mario.mauri@unimib.it (M.M.); v.crippa15@campus.unimib.it (V.C.); ilaria.crespiatico@unimib.it (I.C.); daniele.ramazzotti@unimib.it (D.R.); carlo.gambacorti@unimib.it (C.G.-P.); rocco.piazza@unimib.it (R.P.); luca.mologni@unimib.it (L.M.); 2Department of Pathology, Boston Children’s Hospital and Harvard Medical School, Boston, MA 02115, USA; simone.piane@childrens.harvard.edu; 3Hematology Division, Fondazione IRCCS San Gerardo dei Tintori, 20900 Monza, Italy; federica.cocito@irccs-sangerardo.it (F.C.); elisa.bossi@irccs-sangerardo.it (E.B.); 4Lymphoma Unit, Department of Onco-Hematology, IRCCS San Raffaele Scientific Institute, 20132 Milan, Italy; steidl.carolina@hsr.it; 5Experimental Immunology Unit, IRCCS San Raffaele Scientific Institute, 20132 Milan, Italy; ivancivettini@gmail.com; 6Transfusion Medicine Unit, Fondazione IRCCS San Gerardo dei Tintori, 20900 Monza, Italy; chiaramariadele.scollo@irccs-sangerardo.it

**Keywords:** ALK, macrophages, tumor immunology, CD47, TKIs (tyrosine kinase inhibitors), neuroblastoma, lymphoma, NSCLC, tumor microenvironment (TME)

## Abstract

Background: Anaplastic lymphoma kinase (ALK) plays a role in the development of lymphoma, lung cancer and neuroblastoma. While tyrosine kinase inhibitors (TKIs) have improved treatment outcomes, relapse remains a challenge due to on-target mutations and off-target resistance mechanisms. ALK-positive (ALK+) tumors can evade the immune system, partly through tumor-associated macrophages (TAMs) that facilitate immune escape. Cancer cells use “don’t eat me” signals (DEMs), such as CD47, to resist TAMs-mediated phagocytosis. TKIs may upregulate pro-phagocytic stimuli (i.e., calreticulin, CALR), suggesting a potential therapeutic benefit in combining TKIs with an anti-CD47 monoclonal antibody (mAb). However, the impact of this combination on both TKIs-sensitive and resistant ALK+ tumors requires further investigation. Methods: A panel of TKIs-sensitive and resistant ALK+ cancer subtypes was assessed for CALR and CD47 expression over time using flow cytometry. Flow cytometry co-culture and fluorescent microscopy assays were employed to evaluate phagocytosis under various treatment conditions. Results: ALK inhibitors increased CALR expression in both TKIs-sensitive and off-target resistant ALK+ cancer cells. Prolonged TKIs exposure also led to CD47 upregulation. The combination of ALK inhibitors and anti-CD47 mAb significantly enhanced phagocytosis compared to anti-CD47 alone, as confirmed by flow cytometry and fluorescent microscopy. Conclusions: Anti-CD47 mAb can quench DEMs while exposing pro-phagocytic signals, promoting tumor cell phagocytosis. ALK inhibitors induced immunogenic cell damage by upregulating CALR in both sensitive and off-target resistant tumors. Continuous TKIs exposure in off-target resistant settings also resulted in the upregulation of CD47 over time. Combining TKIs with a CD47 blockade may offer therapeutic benefits in ALK+ cancers, especially in overcoming off-target resistance where TKIs alone are less effective.

## 1. Introduction

Cancer cells employ complex strategies to evade the immune system, allowing them to escape innate and adaptive surveillance. In recent years, the landscape of cancer research and treatment has undergone a transformative shift with the advent of immunotherapy [1].

At the forefront of innate immunity, targeting tumor-associated macrophages (TAMs) within the tumor microenvironment (TME) has been considered a promising immunotherapeutic strategy [2,3]. In fact, several mechanisms have been identified as mediators able to dampen TAMs activity, particularly by blocking phagocytosis [2]. These pathways, also known as “don’t eat me” signals (DEMs), have been validated in both solid and blood malignancies, fostering immune evasion and cancer survival [3]. CD47 was the first DEM recognized. Other pathways have been discovered throughout the years, such as the programmed cell death ligand 1 (PD-L1), the major histocompatibility class I complex (MHC-I), CD24, stanniocalcin-1 (STC-1) and the disialoganglioside GD2 [4,5,6,7,8,9,10].

More specifically, CD47 serves as a DEM on tumor cells by escaping phagocytic elimination mediated by TAMs and other phagocytes through its interaction with signal regulatory protein-α (SIRP-α), which is expressed on macrophages [3]. Although CD47 is mainly upregulated by cancer cells, it is also expressed on many normal cells [11]. The broad distribution of CD47 on normal cells raises concerns about potential toxicity when contemplating a therapeutic strategy targeting CD47. Intriguingly, in a pre-clinical study, the anti-CD47 monoclonal antibody (mAb) demonstrated the ability to eliminate leukemic cells through phagocytosis while sparing normal cells [4]. The success of therapeutic CD47 targeting relies not solely on countering the anti-phagocytic CD47 signal, but also on considering the presence of pro-phagocytic or “eat me” stimuli on a cell that needs to be phagocytosed: when normal cells undergo damage, they promote the expression of pro-phagocytic molecules, thus facilitating their homeostatic removal through phagocytosis [12]. Consequently, cancer cells express both CD47 and pro-phagocytic signals, whereas healthy normal cells only express CD47. In a therapeutic context, anti-CD47 mAb may inhibit CD47 DEMs, thus inducing phagocytosis of cancer cells expressing pro-phagocytic molecules [13,14]. In contrast, blocking CD47 on normal cells would trigger no phagocytosis due to the absence of positive “eat me” signals [13].

In this scenario, calreticulin (CALR) has been identified as one of the predominant “eat me” signals operating in conjunction with CD47 [14]. CALR is typically localized in the endoplasmic reticulum and undergoes translocation to the cell surface in response to cellular damage or stress [15]. It functions as a pro-phagocytic signal, identifying cells for clearance: in fact, CALR forms a complex with its ligand, low-density lipoprotein-related protein (LRP), which is present on macrophages, thereby initiating the engulfment of targeted cells [12]. Although CALR can be induced on cancer cells, macrophages can also release CALR, which binds to cancer cells and triggers macrophage phagocytosis [14,16]. In response to this pro-phagocytic stimulus, cancer cells may upregulate CD47 as a protective mechanism to evade both phagocytosis and the immune system [14].

Therefore, the interplay between CD47 and CALR underscores their integral roles in modulating immune responses and presents promising avenues for therapeutic strategies, which should be oriented to selectively upregulate CALR and quench CD47 signaling on cancer cells.

Recently, Petrazzuolo and colleagues showed that anaplastic lymphoma kinase (ALK) inhibition causes immunogenic cell damage through the upregulation of CALR in a panel of ALK-driven lymphoma cell lines [17]. ALK is an oncogene involved in the pathogenesis of various cancers [18,19]. In fact, ALK rearrangements are a hallmark of anaplastic large cell lymphoma (ALCL) and have also been identified in a subset of non-small cell lung cancer (NSCLC) [19]. Additionally, the mutated full-length ALK receptor is a driver of neuroblastoma, and a high expression of ALK is associated with a dismal prognosis [20,21]. While the use of tyrosine kinase inhibitors (TKIs) represents a significant advancement in the therapeutic management of these tumors, providing more specific and effective treatment options, resistance in tumor cells may occur during the use of TKIs, which is partially explained by the occurrence of mutations in the ALK domain that halts the drug interaction with the oncogene [19,22,23,24,25,26,27,28]. As a matter of fact, alternative resistance may occur through off-target mechanisms, involving the activation of bypass signals independent of ALK, such as the activation of phosphoinositide 3-kinase (PI3K)/AKT (PI3K/AKT) and RAS/mitogen-activated protein kinase (RAS/MAPK) pathways, as well as the downregulation of phosphatases like PTPN1 and PTPN2 [29,30,31].

Moreover, ALK protein was found to be immunogenic in humans, and patients with ALK-driven lymphoma and NSCLC are known to spontaneously develop a natural immune response against ALK protein [32,33,34,35,36]. In fact, antibodies against ALK were found in patients with ALK-rearranged ALCL and NSCLC, and a CD8^+^ cytotoxic T lymphocyte (CTL)-mediated response to ALK peptides was documented in peripheral blood lymphocytes (PBLs) from healthy donors [32,33,34]. In addition, an immunogenic response against ALK was associated with a reduced risk of dissemination and relapse in the setting of ALK-driven ALCL [36].

The capability of ALK to induce an adaptive immune response in ALK-driven cancers could be integrated with the restoration of the innate immune system by blocking CD47-mediated DEMs within the TME. In fact, it was demonstrated that enhanced phagocytosis induced by anti-CD47 treatment could improve the surrounding adaptive immunity by priming a specific CD8^+^ CTL response with an effective cytotoxic function, which was secondary to the improved antigen-presenting activity of TAMs after an increase in phagocytosis [37].

The combination between ALKi and anti-CD47 mAb was first explored by Vaccaro et al., demonstrating that targeted therapies can enhance phagocytosis through CD47 blockade in lung cancer [38]. However, further studies are needed to evaluate the effectiveness of this approach across other ALK+ cancer subtypes, particularly in resistant cases where conventional treatment often fails.

## 2. Materials and Methods

### 2.1. Chemicals and Cell Culture

Lorlatinib and crizotinib were provided by Pfizer, and alectinib was purchased from Selleck Chemicals. Karpas-299 (K299) and SUP-M2 cell lines were purchased from DSMZ (Braunschweig, Germany). An AS4 cell line was obtained from ex vivo K299 xenograft, as explained elsewhere [30]. An H3122 cell line was provided by Dr. Claudia Voena (University of Turin, Turin, Italy). CLB-Ga cells were provided by Dr. Valérie Combaret (Léon Bérard Cancer Centre, Lyon, France). All cell lines were cultured in Roswell Park Memorial Institute (RPMI) 1640 (Euroclone, Milan, Italy) with 10% fetal bovine serum (FBS; Euroclone, Milan, Italy), 1 mM L-glutamine and 100 U/mL penicillin/streptomycin (Euroclone, Milan, Italy), in a humidified, 5% CO_2_ incubator at 37 °C. ALK inhibitors were replenished every 48 h during cell culturing, along with the addition of fresh medium. The completed characteristics of cell lines’ resistance to ALK inhibitors are depicted in Appendix A. All cell lines were not independently authenticated beyond the identity provided by the institution of origin and were routinely tested for mycoplasma contamination.

As pertains to the drug concentrations used for all the in vitro analyses, we considered, for lymphoma cell lines, crizotinib at a concentration of 0.5 µM and lorlatinib at 0.1 µM, deliberately excluding higher concentrations of TKIs due to their propensity to increase the number of apoptotic and dead cells, consequently triggering phagocytosis aberrantly [7,17]. Similarly, when examining CALR and CD47 expression over time, we opted for reduced TKIs concentrations (i.e., crizotinib 120 nM and lorlatinib 100 nM), as higher levels provided intolerable toxicity for long-term exposure analysis. For solid malignancies, adhering to the standard of care in clinical settings, we treated H3122 cell line with alectinib at 2 µM and 200 nM for 20 h and long-term exposure analyses, respectively. Lorlatinib was used to treat off-target resistant H3122 with hyperactivated epidermal growth factor receptor (EGFR) to further extend experimental analysis in the resistant setting. Neuroblastoma cell line was treated with lorlatinib at 1 µM; specifically, the lorlatinib-sensitive CLB-Ga cell line was excluded for in vitro analyses due to its high susceptibility to TKIs exposure. Conversely, the lorlatinib-resistant CLB-Ga-LR1000 cell line was employed for both 20 h and long-term analyses, without experiencing toxicity concerns at higher lorlatinib concentrations, given its resistant nature. Extensive characterization of the TKIs-resistant cancer cell lines used was previously undertaken [30].

### 2.2. Antibodies and Reagents

Unconjugated human anti-CD47 mAb (clone B6H12.2) and human IgG_1_ isotype control were purchased from BioXCell (Lebanon, NH, USA) and stored at 4 °C. Phycoerythrin (PE)-conjugated human anti-CD47 (clone REA220), PE-conjugated human anti-CD11b (clone REA713), FITC-conjugated human anti-CD14 (clone REA599) and FITC-conjugated human anti-SIRP-α (clone REA144) mAbs were purchased from Miltenyi Biotec (Bergisch Gladbach, Germany), whereas PE-conjugated human anti-calreticulin (clone FMC 75) was purchased from Abcam (Cambridge, UK) and kept at 4 °C. Recombinant human granulocyte-macrophage colony-stimulating factor (GM-CSF), interleukin-10 (IL-10), transforming growth factor-β1 (TGF-β1) and interferon-γ (IFN-γ) were bought from PeproTech (Rocky Hill, NJ, USA). Lipopolysaccharides compound (LPS) was bought from Sigma Aldrich (Burlington, MA, USA) and kept at −20 °C. Additional details are described in Appendix A.

### 2.3. Monocyte Isolation and Macrophage Differentiation

As previously shown [10], monocytes were purified from peripheral blood mononucleated cells (PBMCs), which were collected from the blood of healthy donors. Blood was diluted at a 1:2 ratio with phosphate-buffered saline (PBS) and separated on Lympholyte-H (Cedarlane, Burlington, ON, Canada), using centrifugation to obtain PBMCs. Monocytes were then isolated from PBMCs using a CD14 Microbeads isolation kit (Miltenyi Biotech, Bergisch Gladbach, Germany), following the manufacturer’s protocols. Isolated monocytes were then checked for purity using flow cytometry (CD14^+^ cells > 90%). For macrophage differentiation, isolated monocytes were cultured with Iscove’s Modified Dulbecco’s Medium (IMDM, Thermo Fisher, Waltham, MA, USA) + 10% AB human serum from Sigma Aldrich for 5–6 days, at a density of 1.5 × 10^5^ cells/cm^2^, in a humidified 5% CO_2_ incubator at 37 °C. At day 0, 50 ng/mL GM-CSF was added to provide differentiation, and unstimulated (M0) macrophages received only this cytokine. Conversely, M2-like phenotype was obtained by adopting 50 ng/mL human IL-10 and 50 ng/mL human TGF-β_1_ on days 3–4 of differentiation until use on days 7–9 [7], whereas 20 ng/mL human IFN-γ and 50 ng/mL LPS on days 5–6 were used to induce M1-like differentiation [10]. M1-like and M2-like macrophages were produced, in order to compare them with M0 macrophages in terms of SIRP-α expression.

### 2.4. Flow Cytometry Analysis

Tumor cells were collected using centrifugation (suspended cells) or detachment with 2 mM ethylenediaminetetraacetic acid (EDTA, adherent cells) before being washed twice with fluorescence-activated cell sorting (FACS) buffer (PBS with 2% FBS) and thereafter stained with PE-conjugated human anti-CD47 mAb (clone REA220) or PE-conjugated human anti-calreticulin (clone FMC 75) for 20 min in the dark at 4 °C. Cells were finally stained with 7-AAD (Miltenyi Biotech) for viability for 5 min in the dark at room temperature. The analysis of CD47 expression over time was not possible for SUP-M2, due to the high susceptibility of the cell line to TKIs and the subsequent high number of dead cells observed during analysis. Anti-CD14 mAb was used to check the purity of isolated monocytes, and anti-SIRP-α mAb (clone REA144) was employed to check SIRP-α expression on M0/M1/M2 macrophages. Samples were analyzed using Attune™ NxT Flow Cytometer (Thermo Fisher) and data were interpreted using FCS Express™ version 7 (De Novo Software, Pasadena, CA, USA).

### 2.5. PCR, Proliferation and Apoptosis Assays

To assess any changes in CD47 RNA expression after ALKi exposure over time, we performed real-time quantitative polymerase chain reaction (qPCR). Briefly, total RNA was extracted using TRIzol (Invitrogen, Carlsbad, CA, USA. Thermo Fisher), and TaqMan qPCR for CD47 (primer: Hs00179953_m1) was performed using Brilliant-II QPCR Master Mix (Agilent, Santa Clara, CA, USA) and probe mixes from Thermo Fisher. The beta-glucuronidase (GUS) gene was used as a reference (probe 50-CCAGCACTCTCGTCGGTGACTGTTCA-30). To define the effects of ALKi to cancer cell proliferation at different timepoints, cells (10.000/well) were treated with the experimental drug for 20 and 72 h. Cell growth was measured with the CellTiter 96 AQueous One Solution Cell Proliferation Assay System (Promega, Madison, WI, USA). Dose–response curves were produced using GraphPad Prism 10, and the IC_50_ value was defined as the concentration of the drug able to inhibit 50% of the vehicle-treated control response (absolute IC_50_).

For cancer cells apoptosis and dead cell quantification, after 20 h of exposure to ALKi, the eBioscience Annexin V Apoptosis Detection Kit PE (Thermo Fisher Scientific) with 7-AAD was employed and analyzed using flow cytometry, according to the manufacturer’s protocol.

### 2.6. Flow Cytometry-Based Phagocytosis Assay

As previously reported [7,10], in vitro phagocytosis was based on co-culture in serum-free IMDM of target tumoral cells and effector cells (i.e., M0 macrophages), at a ratio of 1:2 (target cells: 100.000; effector cells: 50.000), for 1–2 h, in a humidified 5% CO_2_ incubator at 37 °C within ultra-low-attachment 96-well U-bottom plates (Corning, Corning, NY, USA). Briefly, macrophages were collected after detachment with TrypLE Express (Life Technologies, Carlsbad, CA, USA) and gentle scraping and finally resuspended with serum-free IMDM. Cancer cell lines were collected after centrifugation (i.e., suspended cells: K299, SUP-M2, AS4) or 2 mM EDTA detachment (i.e., adherent cells: H3122, CLB-Ga) and subsequently stained with carboxyfluorescein succinimidyl ester (CFSE, Thermo Fisher). After washing with PBS, tumoral cells were resuspended at a concentration of 20 × 10^6^/mL and labeled with 10 μM of CFSE for 10 min in the dark at 37 °C; the reaction was then stopped with RPMI 1640, supplemented with 10% FBS, for 5 min in the dark at room temperature; cells were finally washed twice with PBS and resuspended with serum-free IMDM.

Phagocytosis assays were conducted using the anti-CD47 monoclonal antibody (clone B6H12.2, BioXCell) and an IgG1 isotype control, each at a concentration of 10 μg/mL. The phagocytosis reaction was halted by placing the samples on ice. Following this, the cell suspension was washed with ice-cold PBS and incubated with PE-labeled anti-CD11 antibody (clone REA 713, Miltenyi Biotec) for 20 min in the dark at 4 °C to label human macrophages. After another wash with ice-cold PBS, cells were stained with 7-AAD to assess viability prior to analysis. Samples were processed on an Attune™ NxT Flow Cytometer (Thermo Fisher), and data were analyzed using FCS Express™ (De Novo Software). Phagocytosis was quantified by calculating the percentage of CD11b^+^/CFSE^+^ macrophages among the total CD11b^+^ macrophage population. Each phagocytosis assay was performed in technical triplicates, with normalization to the highest technical replicate per donor due to the variability in phagocytic activity across donor-derived macrophages, as described previously [7,10]. Normalized phagocytosis was then expressed as fold change values; that is, the ratio between the normalized phagocytosis in the “anti-CD47 mAb” condition and the mean value of normalized phagocytosis found in the “IgG_1_ isotype control” condition. Fold change values of normalized phagocytosis were finally considered to compare different co-culture conditions related to the same cell line, in order to check any increase in phagocytosis, particularly in the cases of multiple combinations (i.e., ALKi ± anti-CD47 mAb).

### 2.7. Fluorescent Phagocytosis Microscopy

For fluorescent microscopy co-culture assays, the AS4 cell line was labeled with pHrodo™ Red succinimidyl ester (pHrodo™ Red, SE) (Thermo Fisher), and macrophages were stained with Hoechst 33342 (Thermo Fisher) according to the manufacturer’s protocol [7,39]. Cell suspension of 500.000 AS4 cells, previously exposed to ALKi, and 250.000 macrophages were co-cultured with anti-CD47 mAb at a concentration of 10 μg/mL and incubated in a serum-free IMDM within a 4-well Nunc™ Lab-Tek™ II Chamber Slide™ System (Thermofisher) for 1–2 hours, in a humidified 5% CO_2_ incubator at 37 °C. Following incubation, chambers were placed on ice and thoroughly washed with ice-cold PBS to remove any non-phagocytosed AS4 cells. Each chamber was then examined using an inverted fluorescent microscope (Zeiss AxioObserver, Zeiss, Oberkochen, Germany), and images were processed using ImageJ software version 1.53c (NIH, Bethesda, MD, USA). The phagocytic index was calculated by counting the number of red-labeled tumor cells within macrophages, expressed as the number of ingested cells per 100 macrophages. A minimum of 200 macrophages was counted per condition to ensure reliable quantification.

### 2.8. Statistics

Statistical differences between groups were assessed using the unpaired two-tailed Mann–Whitney U test or Student’s *t*-test. For comparisons among multiple groups, a one-way or two-way Analysis of Variance (ANOVA) was applied to determine statistical significance. A *p*-value < 0.05 was considered statistically significant, with results reported as mean ± SD. Significance levels were indicated as follows: * *p* < 0.05, ** *p* < 0.01, *** *p* < 0.001, **** *p* < 0.0001. All analyses were performed using GraphPad Prism 10.

## 3. Results

### 3.1. Upregulation of CALR and CD47 After Exposure to TKIs in ALK-Positive Cell Lines

We investigated whether CALR surface expression was upregulated after exposure to fixed concentrations of TKIs in a panel of ALK-positive cancer cell lines available in our institution. The upregulation of CALR was assessed using flow cytometry (FC) after exposing cell lines to TKIs for 20 h, as previously shown [17]. An increased CALR^+^/7-AAD^−^ cell population was documented in sensitive ALK-positive ALCL cell lines (K299, SUP-M2) after the administration of crizotinib and lorlatinib (0.5 µM and 0.1 µM, respectively), when compared to the untreated conditions (Figure 1A,B). Conversely, AS4 is an ALK-positive ALCL cell line that presented on-target and off-target resistance to crizotinib and lorlatinib, respectively (Appendix A). In this setting, crizotinib was not effective in upregulating CALR, due to the gatekeeper mutation impeding drug binding, whereas lorlatinib induced immunogenic damage with an overexpression of CALR (Figure 1B). Similarly, ALK-driven solid cancer cell lines experienced CALR upregulation after TKIs exposure. The sensitive and lorlatinib-resistant lung adenocarcinoma H3122 cell line showed an increase in CALR after 20 h of alectinib and lorlatinib, respectively (alectinib 2.0 µM, lorlatinib 0.1 µM; Figure 1C, left and middle panels). The prolonged administration of low-dose TKIs (alectinib 200 nM, lorlatinib 0.1 µM, until day +8) was associated with a trend of CALR downregulation compared to CALR levels after 20 h exposure (Figure 1C, left and middle panels).

The off-target lorlatinib-resistant neuroblastoma cell line (CLB-Ga-LR1000) also experienced the upregulation of CALR after incubation with lorlatinib (1.0 µM) for 20 h (Figure 1C, right panel). In agreement with H3122 findings, the downregulation of CALR was documented after long exposure to lorlatinib at day +8 (Figure 1C, right panel). CALR expression over time was analyzed in ALK-positive lymphoma K299 cell line, showing that, even in this case, surface levels of CALR were reduced after long exposure to low-dose crizotinib (i.e., 120 nM), when compared to the untreated condition (Figure 1D). We also documented that the overexpression of CALR, found at the TKIs concentrations used, was not associated with a reduction in cell proliferation or a significant increase in the number of dead and apoptotic cells (Appendix A).

To finally conclude our investigation in the context of long-term TKIs incubation, we analyzed CD47 expression in our ALK-driven cell lines over time using flow cytometry. We found that lymphoma cell line K299 showed upregulation of CD47 after being exposed for several days to crizotinib (120 nM), reaching higher CD47 levels than the untreated condition at day +11 (Figure 2A).

Conversely, the AS4 cell line, whose resistance against crizotinib is mediated by on-target mechanisms, failed to show evidence of CD47 upregulation (Figure 2B), whereas lorlatinib (100 nM) caused increased CD47 expression (Figure 2C). CD47 upregulation was also documented in solid cancer cell lines, after the extended administration of alectinib (200 nM; for sensitive H3122), lorlatinib (100 nM; for off-target resistant H3122-LR100) and lorlatinib (1000 nM; for off-target resistant CLB-Ga-LR1000; Figure 2D–F). Similarly to AS4, off-target resistance in CLB-Ga-LR1000 and H3122-LR100 did not affect lorlatinib activity in terms of CD47 upregulation (Figure 2E). The overexpression of CD47 by FC was also associated with increased transcriptional CD47 RNA levels, as shown using qPCR (Appendix A).

Altogether, in both sensitive and resistant settings, these data indicate that short-term, fixed-dose TKIs administration resulted in the upregulation of CALR, whereas long-term, low-dose TKIs incubation was associated with the downregulation of CALR and the overexpression of CD47 levels.

### 3.2. Enhanced Phagocytosis After Combination of TKIs and CD47.DEMs Blockade in Sensitive and Resistant ALK-Driven Malignancies

The upregulation of CALR and CD47 found after short and long-term TKIs incubation, respectively, was considered to enhance phagocytosis with CD47.DEMs blockade, thus defining a new therapeutic strategy in ALK-positive malignancies. Therefore, as elsewhere shown, co-culture assays were deployed using M0 macrophages, previously differentiated from isolated donor-derived monocytes [4]. M0 macrophages were used as effector cells, since they showed the consistent expression of SIRP-α by FC, whose levels were compared with the M1- and M2-like phenotypes (Appendix A) [10].

As expected, flow cytometry-based assays showed an increase in phagocytosis when TKIs-exposed cell lines were co-cultured with macrophages and anti-CD47 mAb, in both sensitive and resistant settings (Figure 3A–C, Appendix A).

Indeed, in ALK-positive lymphoma sensitive cell lines (K299, SUP-M2), previous 20 h exposure to crizotinib or lorlatinib (i.e., 0.5 µM and 0.1 µM, respectively) induced increased phagocytosis with anti-CD47 mAb (Figure 3A). Conversely, to assess the phagocytic activity in the resistant setting, we used the AS4 cell line, which is known to carry on-target and off-target resistance mechanisms against crizotinib and lorlatinib, respectively (Appendix A). In agreement with the corresponding CALR findings, crizotinib did not provide an increase in phagocytosis when anti-CD47 mAb was adopted (Figure 3B, left panel). In contrast, lorlatinib still played a role, with significant evidence of phagocytosis in combination with anti-CD47 mAb (Figure 3B, left panel). Interestingly, the combined effect of lorlatinib and anti-CD47 mAb was greater after long-term TKIs incubation, likely secondary to the CD47 upregulation found as a late-induced event (Figure 2C and Figure 3B, right panel).

Similarly to the lymphoma subtype, both ALK-driven NSCLC and neuroblastoma, previously incubated with short-term TKIs (20 h of alectinib or lorlatinib for H3122 and 20 h of lorlatinib for CLB-Ga), were more susceptible to phagocytosis with anti-CD47 mAb during co-culture assay (Figure 3C and Appendix A). Intriguingly, probably due to CD47 upregulation, long-term incubation with TKIs provided higher levels of phagocytosis in both solid cancer subtypes, reporting a significant improvement in phagocytosis in the off-target resistant setting as well (i.e., H3122-LR100, CLB-Ga-LR1000; Figure 2D,E and Figure 3C, left and right panel).

### 3.3. Validation of TKIs and Anti-CD47 Phagocytosis as Phagocytic Index in pHrodo Red Fluorescent Microscopy Assay

To further validate the aforementioned findings obtained by FC, we analyzed phagocytosis using fluorescent microscopy, by staining the AS4 cell line with pHrodo Red dye and macrophages with Hoechst 33342. Labeled pHrodo Red AS4 cells turn red once they are engulfed by macrophages and phagosomes are fused with low-pH lysosomes [39]. The number of pHrodo^+^ events within macrophages was counted to obtain the phagocytic index, determined as the number of ingested cells per 100 macrophages (counting at least 200 macrophages per condition) [4,7,10]. The highest number of pHrodo-Red^+^ events was shown when macrophages were co-cultured with anti-CD47 mAb and AS4 cells, previously exposed to lorlatinib; conversely, crizotinib failed to induce any improvement in phagocytosis (Figure 4A,B and Appendix A). In accordance with FC results, fluorescent microscopy assay confirmed that lorlatinib enhanced anti-CD47-mediated phagocytosis, likely secondary to the immunogenic upregulation of CALR, even in the AS4 setting where off-target resistance mechanisms are active (Figure 4A,B).

## 4. Discussion

ALK-driven cancers comprise a panel of blood and solid malignancies that are associated with a dismal prognosis, particularly in the relapsed/refractory setting [18]. In recent years, TKIs against ALK were valid therapeutic options in this field, even though relapse and disease recurrence may occur after this biological regimen [18,19]. Several efforts have been made to identify the basis of mechanisms of resistance to ALKi, initially identifying on-target mutants that halted the efficacy of first- and second-generation TKIs (e.g., crizotinib, alectinib) such as the G1202R mutant [40]. Third-generation TKIs (i.e., lorlatinib) were found to circumvent on-target resistance to previous molecules, providing more potent inhibitory activity [41]. Nevertheless, resistance to lorlatinib also developed, partly caused by off-target mechanisms, i.e., by processes bypassing ALK dependency [30]. As a matter of fact, our group previously discovered activation of RAS/MAPK and PI3K/AKT pathways as off-target resistance mechanisms to lorlatinib in a preclinical model of ALCL (i.e., AS4 cell line). Similarly, we also showed that the NSCLC model acquired hyperactivation of EGFR after exposure to lorlatinib (i.e., H3122-LR100 cell line), whereas the lorlatinib-resistant neuroblastoma cell line GLB-Ga-LR1000 was associated with the hyperactivation of EGFR and ErbB4 [30]. In agreement with the presence of off-target mechanisms able to explain the lack of response to TKIs, the resistance to crizotinib in ALCL was also explained by the concomitant downregulation of phosphatases PTPN1 and PTPN2, subsequently leading to the hyperactivation of SHP2, MAPK and JAK/STAT pathways [31]. Additionally, the C-C motif chemokine receptor 7 (CCR7) was recently discovered as another mechanism involved in TKIs off-target resistance, as it is able to activate PI3K-γ signaling [42].

Nevertheless, combined inhibitors might still induce long-term selective pressure with the onset of new mechanisms of drug resistance, so alternative therapeutic strategies are still warranted in resistant ALK-driven malignancies [18,19,30].

In this setting, immunotherapy could be a feasible approach since ALK peptides are known to be immunogenic and able to elicit a robust adaptive immune response in patients [32,33,34,35,36]. In fact, recent discoveries pointed out the immunological properties of therapeutic strategies against ALK, such as ALK peptide vaccination in NSCLC and chimeric antigen receptor (CAR)-T cell therapy against ALK in neuroblastoma [43,44]. Moreover, our hypothesis considered the possibility of redirecting TAMs activity against ALK-positive tumors, particularly through the restoration of phagocytosis of cancer cells with CD47.DEMs blockade, which would eventually prime CTL through the presentation of ALK peptides [37]. ALKi might enhance anti-CD47-mediated phagocytosis thanks to the induced immunogenic cell damage, potentially in the off-target resistant setting as well, thus providing a further approach to definitely eliminate TKIs-resistant and persistent cancer cells [17].

Therefore, we extensively validated the upregulation of CALR in a panel of ALK-positive cancer cell lines after the administration of fixed doses of TKIs, showing the overexpression of this “pro-eat me” signal at the concentrations used (Figure 1A–D). As expected, cells harboring on-target mutations were not sensitive to TKIs in terms of CALR overexpression (i.e., crizotinib in AS4 cell line; Figure 1D, left panel), whereas off-target resistances did not interfere with CALR increase when the TKIs used were active on ALK domain (i.e., lorlatinib in AS4 cell line, Figure 1B, right panel; lorlatinib in CLB-Ga-LR1000, Figure 1C, right panel). CALR overexpression seemed to be an early stress event associated with TKIs treatment in vitro, and anti-CD47 treatment could increase phagocytosis of TKIs-exposed ALK+ cancer cells. Surprisingly, longitudinal FC analysis showed a decrease in CALR expression in TKIs-treated settings at later timepoints, which is likely due to cancer cell adaptation to chronic stressful stimuli (Figure 1D). In line with these findings, the evidence of CD47 upregulation, after long-term exposure to TKIs, suggested that ALK-driven malignancies might respond to TKIs immunogenic cell damage by increasing CD47 expression in order to counterbalance initial CALR upregulation, preventing the activation of phagocytosis within the TME (Figure 2A–E and Appendix A). Hence, this could be a potential mechanism of resistance to TKIs, in addition to well-known on-target and off-target ones [30]. We hypothesized that the combination of TKIs and anti-CD47 mAb could restore antitumor immunity by enhancing phagocytosis. This enhancement would result from an initial upregulation of CALR, followed by an increase in CD47 surface levels over time. In this scenario, even if CALR expression decreases after its initial upregulation, minimal CALR levels would still be enough to trigger phagocytosis when the elevated CD47 signal (observed after long-term TKIs treatment) is blocked by the anti-CD47 mAb.

Therefore, in this treatment setting, the primary driver of enhanced phagocytosis would be the suppression of upregulated CD47, which significantly affects phagocytic regulation following prolonged TKI exposure.

In line with this rationale, the administration of anti-CD47 mAb induced an increase in phagocytosis when sensitive and off-target resistant ALK-driven cell lines, previously exposed to TKIs, were co-cultured with macrophages (Figure 3 and Figure 4). The highest rates of phagocytosis were evident in those conditions (i.e., AS4, H3122, CLB-Ga-LR1000) where the long-term administration of TKIs was employed, in accordance with the upregulation of CD47 found after at least one week of TKIs incubation (Figure 3A–C).

Nonetheless, phagocytosis was still enhanced when cell lines exposed to short-term TKIs incubation were used, which was likely secondary to CALR upregulation (Figure 3A,B). In fact, short-term TKIs administration (i.e., crizotinib 0.5 μM, lorlatinib 0.1 μM) was sufficient to upregulate CALR, allegedly becoming the main trigger of phagocytosis, since the number of apoptotic and dead cells at these TKIs concentrations was statistically irrelevant (Appendix A). Moreover, anti-CD47 mAb and previous incubation with crizotinib (0.5 μM) did not increase the phagocytosis of AS4 cells, since no CALR overexpression occurred, due to the presence of the L1196M mutation on the ALK domain. Conversely, even though resistances bypassing ALK domain compromised lorlatinib activity against AS4 cells, the association between the CD47.DEMs blockade and the third-generation ALKi turned out to be effective, providing an enhancement of phagocytosis able to eliminate off-target resistant cells (Figure 3B and Figure 4A,B). This provides a new possible therapeutic approach against bypass resistance, which might become more prevalent as more patients undergo sequential multi-TKIs therapy [45].

Although Vaccaro and Colleagues initially demonstrated that targeted therapy could prime macrophage activity in NSCLC [38], limited information exists regarding the effects of TKIs in different ALK+ malignancies. In this work, we provided preclinical evidence supporting the efficacy of combining TKIs with anti-CD47 mAb in both blood and solid ALK-positive cancers. Beyond previous findings, we showed for the first time that this combined treatment is effective not only in TKIs-sensitive setting, but also in resistant ones, particularly in cases with bypass resistance mechanisms, a major challenge in maintaining ALKi efficacy. Furthermore, our study suggests that CD47 upregulation may represent a novel mechanism of resistance, and targeting this upregulation could significantly restore phagocytic activity, facilitating the eradication of resistant ALKi cancer cells. These findings, though encouraging, are fundamentally constrained by the lack of in vivo validation. Rigorous and comprehensive studies in vivo will be therefore essential to confirm our results, in order to establish a solid groundwork for future clinical implementation.

Even though a clinically effective antibody against CD47 is still missing, strong preclinical results have already shown the importance of DEMs in terms of cancer immune evasion [3,4,11,46,47]. Hence, other antibodies against CD47 are under investigation and new efforts to potentiate the CD47-SIRP-α axis have been recently outlined to ameliorate anti-CD47 effectiveness, such as engineering its fragment crystallizable (Fc)-domain to enhance therapeutic activity while minimizing toxicity [11,48].

The increase in phagocytosis found by integrating ALKi and the CD47.DEMs blockade confirmed that targeting TAMs within the TME is a potentially valid strategy to treat ALK-driven malignancies, providing new therapeutic options in a TKIs-resistant setting.

## Figures and Tables

**Figure 1 biomedicines-12-02819-f001:**
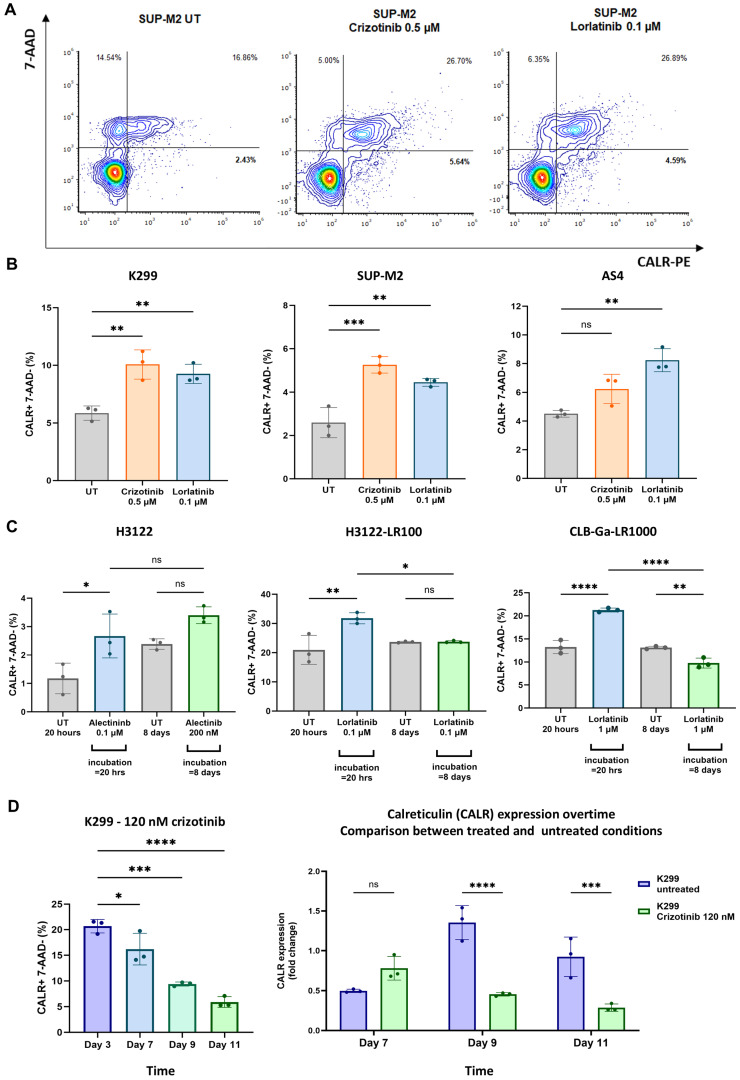
Surface expression of calreticulin (CALR) after tyrosine kinase inhibitors (TKIs) exposure in both sensitive and resistant ALK–positive cancer subtypes. (**A**) Representative flow cytometry plot of CALR+/7–AAD- cell population in SUP-M2 cell line, after exposure to crizotinib (0.5 µM) and lorlatinib (0.1 µM) for 20 h, compared to negative control (untreated, UT). (**B**) Representative histogram bars in terms of percentages of CALR^+^/7–AAD^−^ population for each ALK-positive cancer cell lines available in institution, after exposure to different TKIs for 20 h. (**C**) Expression of CALR in sensitive and lorlatinib–resistant lung adenocarcinoma cell line H3122 after 20 h and 8 days of alectinib and lorlatinib exposure, respectively, (left and middle panels, alectinib 20 h: 2 µM; alectinib 8 days: 200 nM; lorlatinib 20 h and 8 days: 0.1 µM) and CALR expression in resistant neuroblastoma cell line (CLB-Ga-LR1000) after exposure to lorlatinib (1.0 µM) for 20 h and long exposure at day +8 (C, right panel). (**D**) CALR expression over time in K299 cell line after long exposure to crizotinib (120 nM) compared to the untreated setting (one-way ANOVA with multiple comparisons correction; K299 *F*_(2,6)_ = 16.98, SUP-M2 *F*_(2,6)_ = 25.36, AS4 *F*_(2,6)_ = 18.06, H3122 *F*_(2,6)_ = 6.323, H3122-LR100 *F*_(3,8)_ = 9.213, CLB-Ga-LR1000 *F*_(2,6)_ = 92.04, K299 _CRIZO_120 nM_
*F*_(3,8)_ = 42.55, K299 _CALR+DAY7_
*F*_(2,12)_ = 26.39, K299 _CALR+DAY9_
*F*_(2,12)_ = 7.307, K299 _CALR+DAY11_
*F*_(1,12)_ = 35.79; experimental triplicates; ns: not significant; * *p* < 0.05, ** *p* < 0.01, *** *p* < 0.001, **** *p* < 0.0001).

**Figure 2 biomedicines-12-02819-f002:**
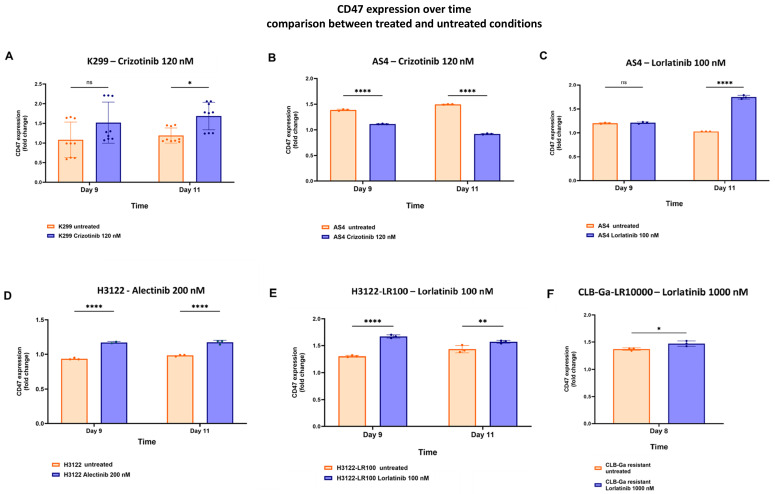
CD47 surface expression after exposure to TKIs in sensitive and off-target resistant ALK-positive cell lines. (**A**–**F**) CD47 surface expression, analyzed using flow cytometry, in lymphoma cell line K299, AS4 cell line, lung cancer cell line H3122 (sensitive to alectinib and resistant to lorlatinib) and neuroblastoma cell line CLB-Ga-LR1000, according to TKIs used; two-way ANOVA with multiple comparisons correction, K299 *F*_(1,32)_ = 12.29, AS4_CRIZO_ *F*_(1,8)_ = 52.08, AS4_LORLA_ *F*_(1,8)_ = 182.8, H3122 *F*_(1,7)_ = 348.5, H3122_LORLA_ *F*_(1,8)_ = 24.54; experimental triplicates; ns: not significant; * *p* < 0.05, ** *p* < 0.01, **** *p* < 0.0001; for CLB-Ga-LR1000: unpaired, one-tailed Student’s *t*-test, * *p* < 0.05).

**Figure 3 biomedicines-12-02819-f003:**
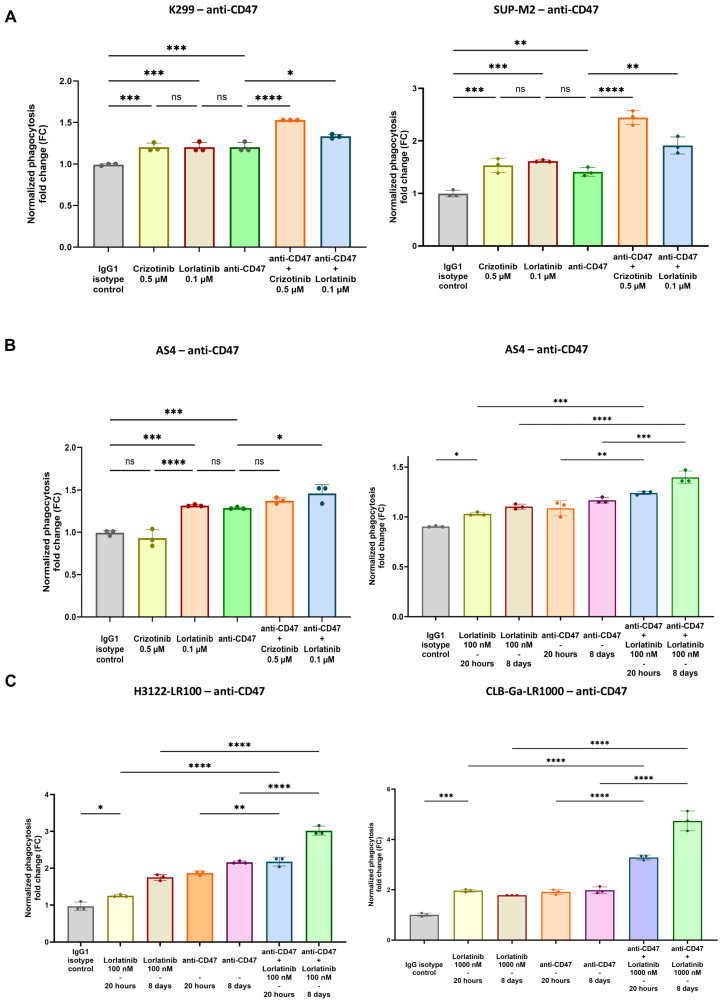
Extended analysis of increased phagocytosis in a panel of ALK-positive tumor cell lines, treated with anti-CD47 mAb, previously exposed to ALKi. (**A**,**B**) Histogram analysis of phagocytic rate in ALK-positive lymphoma sensitive (K299, SUP-M2) and resistant (AS4) setting, previously exposed to crizotinib or lorlatinib, treated with anti-CD47 mAb (**A**); crizotinib 0.5 µM, lorlatinib 0.1 µM). (**C**) Histogram analysis of phagocytic rate in off-target resistant ALK-driven solid cancer cell lines (H3122-LR100, CLB-Ga-LR1000. One-way ANOVA with multiple comparisons correction; K299_CRIZO_ *F*_(3,8)_ = 34.39, K299_LORLA_ *F*_(3,8)_ = 49.50, SUP-M2_CRIZO_ *F*_(3,8)_ = 3.932, SUP-M2_LORLA_ *F*_(3,8)_ = 12.16, AS4_CRIZO_ *F*_(3,8)_ = 8.794, AS4_LORLA_
*F*_(3,8)_ = 9.344, H3122-LR100_LORLA_ *F*_(3,8)_ = 17.07; CLB-Ga-LR1000_LORLA_ *F*_(2,6)_ = 102.2; experimental triplicates, *n* = 3 donors; ns: not significant; * *p* < 0.05, ** *p* < 0.01, *** *p* < 0.001, **** *p* < 0.0001).

**Figure 4 biomedicines-12-02819-f004:**
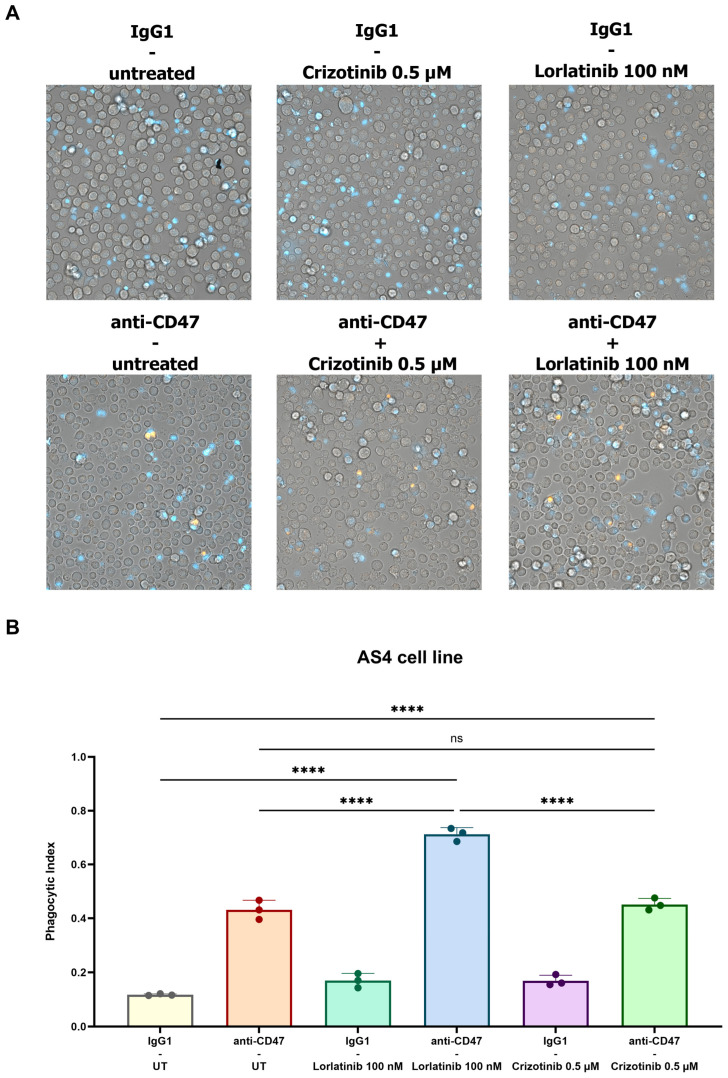
Fluorescent microscopy after incubation of human macrophages with anti-CD47 mAb and the ALK-positive lymphoma AS4 cell line, previously exposed to crizotinib or lorlatinib. (**A**) Representative images of fluorescent microscopy where Hoechst 33342^+^ macrophages were incubated with anti-CD47 mAb and the ALK-positive lymphoma AS4 cell line, labeled with the pH-sensitive dye pHrodo-Red and previously exposed to crizotinib (0.5 µM) or lorlatinib (100 nM) for 20 h. (**B**) Representative histogram bars of phagocytic index (number of pHrodo-red^+^ tumoral cells per 100 macrophages) in case of combination of anti-CD47 mAb and lorlatinib treatment (one-way ANOVA with multiple comparisons correction; AS4 *F*_(5,12)_ = 275.6; technical triplicate; *n* = 1 donor, one experimental cohort; ns: not significant; **** *p* < 0.0001).

## Data Availability

The data that support the findings of this study are available from the corresponding author upon reasonable request.

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
