# Peer review of "Anaplastic Lymphoma Kinase (ALK) Inhibitors Enhance Phagocytosis Induced by CD47 Blockade in Sensitive and Resistant ALK-Driven Malignancies"

_biomedicines, 2024, doi:10.3390/biomedicines12122819_

Round 1
Reviewer 1 Report
Comments and Suggestions for Authors
Malighetti et al. investigated whether combining CD47 blockade with ALK tyrosine kinase inhibitors could be an effective therapeutic strategy for ALK-driven cancers. They tested the effects of ALK inhibition on the expression of the eat-me signal, Calreticulin (CALR), and CD47 in a panel of ALK-positive human cancer cell lines with different sensitivities to ALK inhibition. The results presented indicate that combining ALK inhibition with CD47-blocking antibodies enhances phagocytosis of cancer cells, which confirms findings previously reported in non-small cell lung cancer (Vaccaro et al, JCI, 2024). The authors also provide evidence to suggest this therapeutic combination could be effective in the context of resistance to ALK inhibition, which is an interesting finding that requires further validation. The manuscript requires additional data to substantiate the conclusions drawn, most notably, control data for some experiments, another ALK inhibitor resistant model to understand generalizability, and in vivo treatment studies to robustly demonstrate therapeutic efficacy in ALK inhibitor-resistant cancers.
Major Comments:
1. Since combining CD47 blockade with ALK inhibition has already been shown to enhance phagocytosis of cancer cells (Crizotinib and Lorlatinib) and anti-tumor activity in mice (Lorlatinib) [Vaccaro et al, JCI, 2024], the knowledge advance provided from this study is that this combination could be effective in ALK inhibitor resistant cancers. Therefore, the authors should discuss their results in the context of these previous findings.
2. To confirm that ALK inhibitor resistant cancers are sensitive to the combination proposed, phagocytosis experiments in an additional ALK inhibitor resistant models should be conducted.
3. Because the treatment combination is proposed to activate innate and adaptive immunity to exert its therapeutic effects, in vivo experiments to test efficacy and anti-tumor immune responses in ALK inhibitor resistant models would greatly strengthen the paper.
4. Control data are lacking for some experiments, which limits interpretation of the results. Time-matched controls should be presented for the studies evaluating CD47 and CALR expression (ie. untreated controls for both the 20hr and 8 day time points). IgG isotype control data should be presented in Fig.4 like it was in Fig.5.
Minor Comments:
1. The abstract would benefit from stating a clear objective of the study and explicitly stating the study findings. The end of the introduction would benefit from clearly stating the study objective and describing the general approach taken to address it.
2. The M0, M1, M2 macrophage phenotypes should be confirmed using established markers like CD80/86, MHC-II, CD206, Arg-1, INOS. Has SIRPa been previously shown to have different expression levels on different types of macrophages?
3. If CALR is downregulated after long-term treatment, what eat-me/pro-phagocytic signal is responsible for driving phagocytosis in this treatment context?
4. Suggest merging Fig.3 into Fig.4 or moving it to supplementary material because it is a representative example and the complete data are presented in Fig.4.
5. It is unclear whether CALR and CD47 expression were measured in the same cells or if these were tested in different experiments. If they were measured in separate experiments, a rationale should be provided because the balance of eat-me/don’t eat me signals on a single cell is important for determining whether phagocytosis occurs, and no information is provided about the kinetics of drug-induced changes in CD47 expression (ie. it is only assessed after long-term treatment).
6. Fig.4B – suggest switching the violin plot to a bar plot for consistency with other graphs, and it should include the untreated control group.
Comments on the Quality of English LanguageThe vocabulary used in the manuscript could be improved. Some examples are: line 127 “ALK inhibitors were renovated during cell culture”; line 352 replace “more considerable” with “greater”; line 467 replace “ameliorate” with “augment” or “potentiate”; line 477 “adopted” what does this mean here?
Author Response
Reviewer #1
Malighetti et al. investigated whether combining CD47 blockade with ALK tyrosine kinase inhibitors could be an effective therapeutic strategy for ALK-driven cancers. They tested the effects of ALK inhibition on the expression of the eat-me signal, Calreticulin (CALR), and CD47 in a panel of ALK-positive human cancer cell lines with different sensitivities to ALK inhibition. The results presented indicate that combining ALK inhibition with CD47-blocking antibodies enhances phagocytosis of cancer cells, which confirms findings previously reported in non-small cell lung cancer (Vaccaro et al, JCI, 2024). The authors also provide evidence to suggest this therapeutic combination could be effective in the context of resistance to ALK inhibition, which is an interesting finding that requires further validation. The manuscript requires additional data to substantiate the conclusions drawn, most notably, control data for some experiments, another ALK inhibitor resistant model to understand generalizability, and in vivo treatment studies to robustly demonstrate therapeutic efficacy in ALK inhibitor-resistant cancers.
We would like to thank the Reviewer for the valuable comments. We are aware that, during the final stages of our submission process, a paper on a related topic was published (Vaccaro et al., JCI, 2024). However, our study, conducted over the past three years, presents distinct results and novel findings.
Specifically, we focused on the combination of CD47 "don't eat me" signal-blocking antibodies with TKIs, not only in NSCLC but also in ALK+ lymphoma and neuroblastoma.
Our work also provides new insights into the efficacy of this combination, especially in the setting of ALK-resistant cancers. We found that the combination synergistically counteracts resistance mechanisms mediated by bypass pathways.
Additionally, we identified CD47 upregulation as a novel mechanism of resistance following prolonged TKI exposure, a discovery that has not been previously reported in the context of ALK-targeted TKI treatment.
We fully agree with the Reviewer’s suggestion to cite this paper and make comparisons with our own findings.
Major Comments:
- Since combining CD47 blockade with ALK inhibition has already been shown to enhance phagocytosis of cancer cells (Crizotinib and Lorlatinib) and anti-tumor activity in mice (Lorlatinib) [Vaccaro et al, JCI, 2024], the knowledge advance provided from this study is that this combination could be effective in ALK inhibitor resistant cancers. Therefore, the authors should discuss their results in the context of these previous findings.
We fully agree with the Reviewer and we modified our manuscript to make comparisons between the suggested paper and our own findings.
- To confirm that ALK inhibitor resistant cancers are sensitive to the combination proposed, phagocytosis experiments in an additional ALK inhibitor resistant models should be conducted.
We would like to thank the Reviewer for this consideration. Our lab previously developed ALK+ resistant cancer cell lines with bypass mechanisms of resistance (Redaelli et al, Cancer Res, 2018). Unfortunately, the generation of ALK+ resistant cancer cell lines in vitro is a stochastic process that can take several months, with no certainty that bypass mutation will occur, as resistance could arise solely from mutations within the ALK domain. However, we had the opportunity to test the third and last ALK+ cancer cell line available in Institution (i.e., H3122-LR100), harboring a bypass mechanism of resistance which compromises the sensitivity to lorlatinib. Therefore, we provided co-culture assay with anti-CD47 and TKIs administration. We demonstrated that this third off-target resistant ALK+ cancer cell line was susceptible to the combination of TKI and anti-CD47 in terms of phagocytic rate (Figure 4).
- Because the treatment combination is proposed to activate innate and adaptive immunity to exert its therapeutic effects, in vivo experiments to test efficacy and anti-tumor immune responses in ALK inhibitor resistant models would greatly strengthen the paper.
We would like to thank the Reviewer for this comment. While we value the suggestion, unfortunately, we faced delays in conducting in vivo experiments due to the extended bureaucratic procedures required to obtain authorization for animal experimentation from the Italian Ministry of Health. As the authorization was not granted within a reasonable timeframe, we made the decision to concentrate solely on in vitro analyses for submission to prevent unnecessary delays and to preserve the novelty of the project.
- Control data are lacking for some experiments, which limits interpretation of the results. Time-matched controls should be presented for the studies evaluating CD47 and CALR expression (i.e., untreated controls for both the 20hr and 8day time points). IgG isotype control data should be presented in Fig.4 like it was in Fig.5.
We thank the Reviewer for this comment. We regret any inconvenience caused and have addressed the issue by including time-matched controls where they were previously missing (i.e., Figure 2C, untreated condition at 8 days for H3122 and CLG-Ga-LR1000).
Regarding the co-culture assays, our initial focus was on delineating the distinctions based on the presence or absence of TKIs under the condition of anti-CD47 antibody. To enhance clarity, we have revised the graphs to incorporate the IgG isotype control condition.
Minor Comments:
- The abstract would benefit from stating a clear objective of the study and explicitly stating the study findings. The end of the introduction would benefit from clearly stating the study objective and describing the general approach taken to address it.
We thank the Reviewer for this consideration. We modified the abstract as suggested.
- The M0, M1, M2 macrophage phenotypes should be confirmed using established markers like CD80/86, MHC-II, CD206, Arg-1, INOS. Has SIRPa been previously shown to have different expression levels on different types of macrophages?
We thank the Reviewer for this comment. The phenotype of macrophages was considered out of scope for this paper since it was already outlined in our previous paper (Aroldi et al, J Cell Mol Med, 2023). We agree with the Reviewer’s consideration and cited our manuscript as reference for definition of macrophage phenotyping while comparing SIRP-alpha expression (shown in Figure S5A).
- If CALR is downregulated after long-term treatment, what eat-me/pro-phagocytic signal is responsible for driving phagocytosis in this treatment context?
We would like to thank the Reviewer for this comment. We observed CALR downregulation rather than its complete elimination as pro-eat me signal after long-term treatment; even though CALR levels are low, this minimal expression is enough to trigger phagocytosis when upregulated CD47 signal (found after long-term treatment) is blocked by anti-CD47 mAb. In this setting, enhanced phagocytosis is predominantly attributed to the suppression of the upregulated CD47, which significantly influences phagocytic regulation. We clarified this aspect in the manuscript.
- Suggest merging Fig.3 into Fig.4 or moving it to supplementary material because it is a representative example, and the complete data are presented in Fig.4.
We would like to thank the Reviewer for this suggestion. We moved Fig.3 in the Supplementary material.
- It is unclear whether CALR and CD47 expression were measured in the same cells or if these were tested in different experiments. If they were measured in separate experiments, a rationale should be provided because the balance of eat-me/don’t eat me signals on a single cell is important for determining whether phagocytosis occurs, and no information is provided about the kinetics of drug-induced changes in CD47 expression (ie. it is only assessed after long-term treatment).
We would like to thank the Reviewer for raising this concern. CALR and CD47 were measured in the same cells, so the results derived from the same condition. CD47 expression turns out to increase only after long-time exposure, and the upregulation was monitored over time not only by Flow cytometry (Figure 2), but also by quantitative RT-PCR (Figure S4).
- Fig.4B – suggest switching the violin plot to a bar plot for consistency with other graphs, and it should include the untreated control group.
We thank the Reviewer for this comment. We modified the graph as suggested.
Comments on the Quality of English Language
The vocabulary used in the manuscript could be improved. Some examples are: line 127 “ALK inhibitors were renovated during cell culture”; line 352 replace “more considerable” with “greater”; line 467 replace “ameliorate” with “augment” or “potentiate”; line 477 “adopted” what does this mean here?
We thank the Reviewer for this consideration. We improved the vocabulary in the manuscript as suggested.

Reviewer 2 Report
Comments and Suggestions for Authors
In this manuscript, the authors study immunogenicity confered by ALK expressing cancers, which may be compromised in relapsed/refractory disease. In particular, CD47 vs CALR expression is examined. The authors conclude that CD47 blockade combined with ALK inhibition enhances phagocytosis.
Overall, this is a incremintal study based on previous work and is based on a somewhat obvious premise. Nevertheless, the study is generally well-conducted, and its conclusions are sound.
Comments on the Quality of English LanguageAlthough most of the manuscript is in decent shape, the text in some parts is fairly rough and needs significant work.
Author Response
Reviewer #2
In this manuscript, the authors study immunogenicity confered by ALK expressing cancers, which may be compromised in relapsed/refractory disease. In particular, CD47 vs CALR expression is examined. The authors conclude that CD47 blockade combined with ALK inhibition enhances phagocytosis. Overall, this is a incremintal study based on previous work and is based on a somewhat obvious premise. Nevertheless, the study is generally well-conducted, and its conclusions are sound.
We would like to thank the Reviewer for this comment. While we are aware of a recent publication on a related topic (Vaccaro et al, JCI 2024), our work, conducted over the last three years, focused on the application of the combination of CD47 “don’t eat me” signal blocking antibody with TKIs in different ALK+ cancer subtypes beyond NSCLC, including ALK+ lymphoma and Neuroblastoma. More importantly, our study provided novel insights in terms of efficacy of this combination, particularly in the ALK-resistant setting. In fact, we found that the combination synergistically counteracts bypass mechanisms of resistance.
Moreover, we identified CD47 upregulation as a novel mechanism of resistance following prolonged exposure to TKIs, which has not been previously reported in the setting of TKI treatment against ALK.
Comments on the Quality of English Language
Although most of the manuscript is in decent shape, the text in some parts is fairly rough and needs significant work.
We would like to thank the Reviewer for this observation. We have revised the manuscript to improve its readability and flow.

Round 2
Reviewer 1 Report
Comments and Suggestions for Authors
The manuscript by Malighetti et al. has been improved but still has deficiencies.
1) Controls are still missing in some experiments. There is no data like that in Fig.2S to demonstrate ALK TKI resistance in the H3122-LR100 and CBL0Ga-LR1000 cell lines compared to the parental sensitive lines. There is no phagocytosis data for ALK TKI treatment alone in any of the cell lines presented in Fig.3. This is an essential control to support the authors’ speculation that the combination of TKIs and anti-CD47 mAb “restores” antitumor immunity by enhancing phagocytosis. To support their idea of a combination effect, less anti-tumor immunity (ie phagocytosis) should be induced by TKI alone, but this comparative data is not shown in the resistant models.
2) CALR expression over time is not shown for the newly incorporated H3122-LR100 resistant model (Fig.1).
3) The lack of in vivo experiments for a study relevant to immunotherapy is a major weakness. If the team cannot perform these experiments, they should at least acknowledge this limitation in the discussion.
Author Response
1) Controls are still missing in some experiments. There is no data like that in Fig.2S to demonstrate ALK TKI resistance in the H3122-LR100 and CBL0Ga-LR1000 cell lines compared to the parental sensitive lines. There is no phagocytosis data for ALK TKI treatment alone in any of the cell lines presented in Fig.3. This is an essential control to support the authors’ speculation that the combination of TKIs and anti-CD47 mAb “restores” antitumor immunity by enhancing phagocytosis. To support their idea of a combination effect, less anti-tumor immunity (i.e., phagocytosis) should be induced by TKI alone, but this comparative data is not shown in the resistant models.
In this current manuscript, Figure S2 highlights the differences in IC50 values within the lymphoma setting, serving as a representative example of TKI resistance. Regarding ALK TKI resistance in the H3122-LR1100 and CLB-Ga-LR1000 cell lines, we previously provided a comprehensive characterization of this resistance in a prior manuscript (Redaelli et al., Cancer Res, 2018). In order to avoid further redundancy with previously published data, we clarified in the text that these resistant cell lines have already been characterized and included the corresponding citation in the references section.
For the phagocytosis assays presented in Fig. 3, we included the "TKI alone" condition in all graphs, as recommended.
2) CALR expression over time is not shown for the newly incorporated H3122-LR100 resistant model (Fig.1).
We provided the CALR expression for the newly incorporated H3122-LR100 as suggested.
3) The lack of in vivo experiments for a study relevant to immunotherapy is a major weakness. If the team cannot perform these experiments, they should at least acknowledge this limitation in the discussion.
We fully agree with the Reviewer’s observation and have revised our manuscript to address this point. Specifically, we have expanded the discussion section to acknowledge the limitations arising from the absence of in vivo experiments.
Reviewer 2 Report
Comments and Suggestions for Authors
The author's have addressed my concerns in this revised manuscript.
Author Response
The authors have addressed my concerns in this revised manuscript.
We thank the Reviewer for the feedback.
Round 3
Reviewer 1 Report
Comments and Suggestions for Authors
The authors have adequately addressed the remaining concerns.